# Single Crystal Growth and Superconducting Properties of Antimony-Substituted $NdO_{0.7}F_{0.3}BiS_2$

**Satoshi Demura [1],\*, Satoshi Otsuki [1], Yuita Fujisawa [1], Yoshihiko Takano [2] and Hideaki Sakata [1]**

[1]  Department of Physics, Tokyo University of Science, 1-3 Kagurazaka, Shinjuku-ku, Tokyo 162-8601, Japan; cbe11451@outlook.com (S.O.); 1215702@ed.tus.ac.jp (Y.F.); sakata@rs.kagu.tus.ac.jp (H.S.)

[2]  The World Premier International Center for Materials Nanoarchitectonics (WPI-MANA), National Institute for Materials Science (NIMS), 1-2-1 Sengen, Tsukuba, Ibaraki 305-0047, Japan; takano.yoshihiko@nims.go.jp

\*  Correspondence: demura@rs.tus.ac.jp; Tel.: +81-3-3260-4271 (ext. 2223)

**Abstract:** Antimony (Sb) substitution of less than 8% was examined on a single crystal of a layered superconductor $NdO_{0.7}F_{0.3}BiS_2$. The superconducting transition temperature of the substituted samples decreased as Sb concentration increased. A lattice constant along the *c*-axis showed a large decrease compared with that along the *a*-axis. Since in-plane chemical pressure monotonically decreased as Sb concentration increased, the suppression of the superconductivity is attributed to the decrease in the in-plane chemical pressure.

**Keywords:** $BiS_2$-based superconductor; flux growth; layered structure; superconducting properties; magnetic susceptibility measurement; electrical resistivity measurement

## 1. Introduction

New layered $BiS_2$-based superconductors $Bi_4O_4S_3$ and $Ln(O,F)BiS_2$ ($Ln$ = La, Ce, Pr, Nd, Sm, Yb, and Bi) have been reported [1–9]. Although a superconducting transition temperature ($T_c$) of $Ln(O,F)BiS_2$ is around 4 K, $T_c$ is increased when a lattice strain is introduced. One of the ways to introduce the lattice strain is by partial element substitution with a different ionic radius into $Ln$, Bi, or S site [10–16]. The strain introduced by partial element substitution in these crystals is the chemical pressure. For instance, when selenium (Se) is partially substituted with sulfur (S) in the superconducting layer in $La(O,F)BiS_2$, $T_c$ and a superconducting volume fraction increase [10]. The substitution for a smaller lanthanide ion into the $Ln$ site in the block layer also increases $T_c$ [11–14]. The increase in $T_c$ by element substitution is correlated with the in-plane chemical pressure defined by Mizuguchi: $T_c$ increases as in-plane chemical pressure increases [17]. The change in $T_c$ by the element substitution of $Ln$ or S sites is attributed to the increase in in-plane chemical pressure. On the other hand, it is not known whether the change in $T_c$ is understood in terms of in-plane chemical pressure when an element is partially substituted into the Bi site.

Here, we report on an examination of a substitution of antimony (Sb) ions of less than 8% into $NdO_{0.7}F_{0.3}BiS_2$. Since the Sb ion has the same valence as that of the Bi ion, the effect of the chemical pressure can be investigated independently of the effect of the carrier concentration. We found that the lattice constant along the *c*-axis of Sb-substituted samples decreased. Furthermore, superconductivity was suppressed by the Sb substitution. This suppression by Sb substitution can be explained by the change in the in-plane chemical pressure.

## 2. Experimental Section

$NdO_{0.7}F_{0.3}Bi_{1-x}Sb_xS_2 (x = 0.01–0.08)$ single crystals were synthesized using a CsCl/KCl flux method in an evacuated quartz tubes [18,19]. Powders of $Nd_2S_3$, $Bi_2O_3$, $Bi_2S_3$, $Sb_2S_3$, and $BiF_3$ with

Bi grains were used as a starting material. The $Bi_2S_3$ powders were prepared by reacting Bi and S grains in an evacuated quartz tube at 500 °C for 10 h. A mixture of 0.8 g of starting material and 8 g of CsCl/KCl powder was sealed in an evacuated quartz tube. The tube was heated at 800 °C for 10 h, kept at 800 °C for 10 h, and cooled down to 630 °C at the rate of 0.3 °C/h or 1 °C/h. After this thermal process, the obtained material was washed by distilled water to remove the flux. X-ray diffraction (XRD) patterns using single crystal and powder samples were collected by a Rigaku X-ray diffractometer (Rigaku, Tokyo, Japan) with Cu K$\alpha$ radiation using a $\theta$–$2\theta$ method. These powder samples used in XRD measurements were prepared by grinding single crystals. Lattice constants along the *a*- and *c*-axes were determined using the $\theta$–$2\theta$ technique. The surface condition of the single crystals was observed by a scanning electron microscope (SEM) (JEOL, Tokyo, Japan). The actual chemical composition was determined by an energy-dispersive X-ray (EDX) measurement (JEOL, Tokyo, Japan). Temperature dependence of the magnetic susceptibility down to 2 K was measured with the MPMS (magnetic property measurement system) (Quantum Design, San Diego, CA, USA). Temperature dependence of the electrical resistivity was measured down to 2.5 K with the four terminals method.

## 3. Results

Figure 1a shows X-ray diffraction patterns for $NdO_{0.7}F_{0.3}Bi_{1-x}Sb_xS_2$ (*x* = 0.01–0.08). All peaks correspond to the (00*l*) peaks of the $CeOBiS_2$ type structure with the space group $P4/nmm$ symmetry. The (004) peaks of these samples are magnified in Figure 1b. The peaks are gradually shifted to the high angle side as the Sb concentration increased. This is indicative of success in Sb substitution into $NdO_{0.7}F_{0.3}BiS_2$ up to *x* = 0.08. Figure 1c shows an Sb concentration dependence on the lattice constant along the *c*-axis. The lattice constant is almost constant up to *x* = 0.04 and decreases when *x* > 0.04. A lattice constant along the *a*-axis shows almost constant until *x* = 0.08, as depicted in Figure 1d. The lattice shrinkage in the *c*-axis by Sb substitution was caused by the difference in ionic radius between the Bi and Sb ions, as we expected.

To evaluate sample quality, SEM and EDX measurements were performed in single crystals where *x* = 0.01 and 0.08, as shown in Figure 2. Figure 2a,b show SEM images of these obtained single crystals where *x* = 0.01 and 0.08, respectively. A square shape reflected from the tetragonal crystal structure can be seen. The shape did not change as the Sb concentration increased. In addition, the distribution of all elements included in the crystal was evaluated for the sample where *x* = 0.08 in Figure 2c–h. All elements were homogeneously distributed in the sample. Therefore, the Sb ion was homogeneously substituted into the single crystal of $NdO_{0.7}F_{0.3}BiS_2$ until *x* = 0.08. Furthermore, an actual Sb concentration was estimated for samples where *x* = 0.01 and 0.08. The actual Sb concentrations of the samples with a nominal composition of 0.01 and 0.08 were around 0.001 and 0.021, respectively. The actual composition was quite lower than the nominal composition. Therefore, the actual Sb concentration increased until around 0.02 as nominal Sb concentration increased until 0.08. Next, the superconducting properties of these samples were evaluated to investigate the effect of Sb substitution on superconductivity.

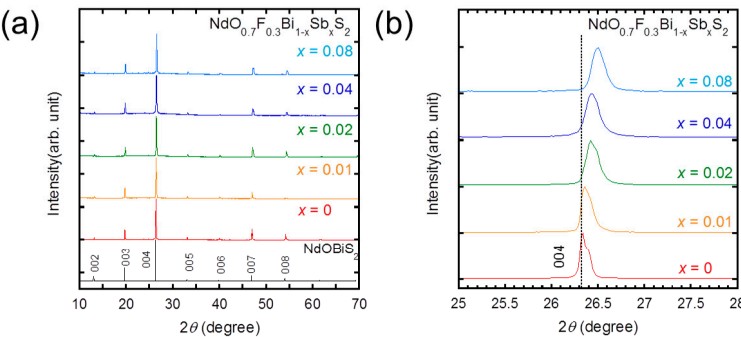

**Figure 1.** *Cont.*

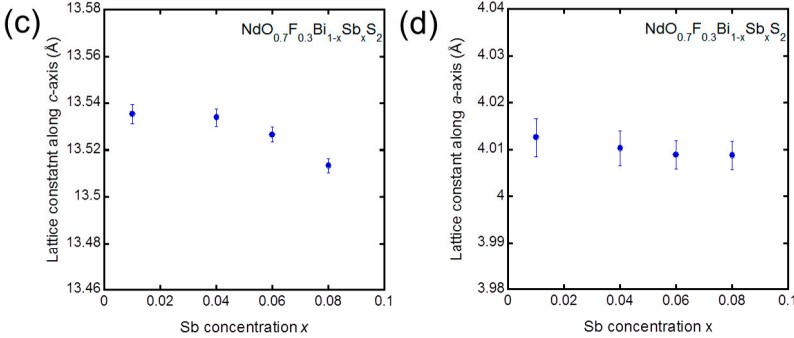

**Figure 1.** XRD patterns and lattice parameters of $NdO_{0.7}F_{0.3}Bi_{1-x}Sb_xS_2$ single crystals. (**a**) XRD patterns from $x = 0.01$ to $0.08$ on the $NdO_{0.7}F_{0.3}Bi_{1-x}Sb_xS_2$. (**b**) Magnified figure near the (004) peak in Figure 1a. (**c**,**d**) Lattice constants along the $c$- and $a$-axes for $NdO_{0.7}F_{0.3}Bi_{1-x}Sb_xS_2$.

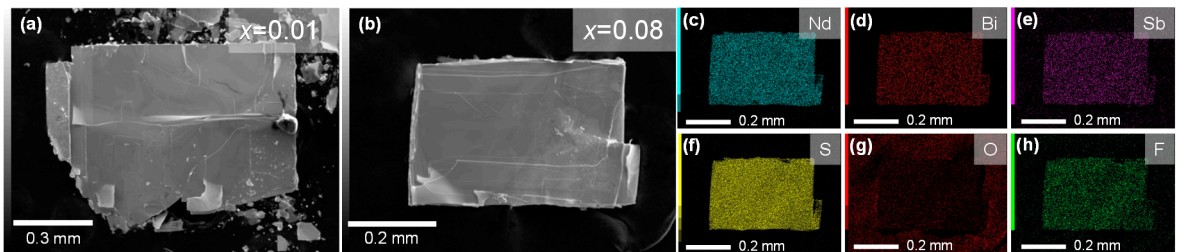

**Figure 2.** (**a**,**b**) SEM images of $NdO_{0.7}F_{0.3}Bi_{1-x}Sb_xS_2$ single crystals ($x = 0.01$, $0.08$). (**c**–**h**) Mapping of the chemical composition of Nd, Bi, Sb, S, O, and F for the single crystal of $NdO_{0.7}F_{0.3}Bi_{1-x}Sb_xS_2$ ($x = 0.08$).

Figure 3a,b show the magnetic susceptibility as a function of temperature for $NdO_{0.7}F_{0.3}Bi_{1-x}Sb_xS_2$ ($x = 0.01$–$0.08$) at a magnetic field of 10 Oe. All samples show diamagnetic signal due to an appearance of superconductivity. A magnitude of $4\pi\chi$ decreases from 1 at $x = 0.01$ to 0.2 at $x = 0.08$. The superconducting transition temperature ($T_c$) monotonously decreases with increasing Sb concentration as shown in Figure 3b. In the zero field cooling process, all samples show a broad superconducting transition. This broad transition is often observed in the other $Ln(O,F)BiCh_2$ ($Ln$ = La, Ce, Pr, Nd, $Ch$ = S, Se) [20–24]. Although the reason why the transition is broadened is not yet understood, the broad transition seems to be a common feature in these materials.

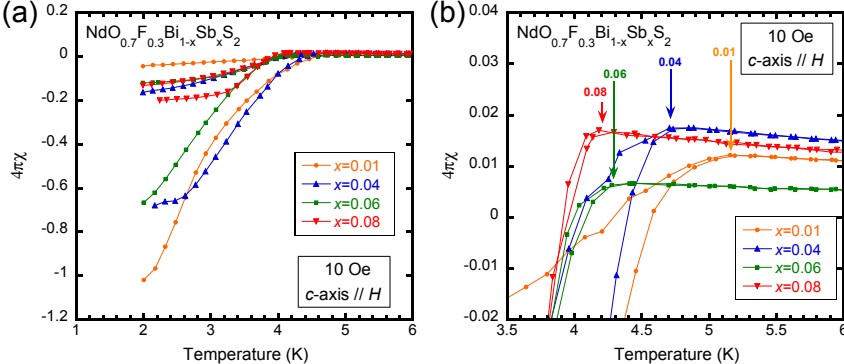

**Figure 3.** Magnetic susceptibility for $NdO_{0.7}F_{0.3}Bi_{1-x}Sb_xS_2$ as a function of temperature. (**a**) Temperature dependence of magnetic susceptibility for $NdO_{0.7}F_{0.3}Bi_{1-x}Sb_xS_2$ from 1 to 6 K. (**b**) The magnetic susceptibility near the superconducting transition for $NdO_{0.7}F_{0.3}Bi_{1-x}Sb_xS_2$.

Figure 4b shows the electrical resistivity near $T_c$ as a function of temperature. A steep drop due to superconductivity can be observed at around 5.2 K in the sample where $x = 0.01$. The transition temperature decreased as the Sb concentration increased. Resistivity at room temperature (RT) continuously increased as the Sb substitution increased, as shown in Figure 4a. This is because the Sb ion was substituted for the Bi ion located in the conduction layers. This increase in resistivity at RT has also been observed in Pb-substituted $Ln$(O,F)BiS$_2$ ($Ln$ = La, Nd) [15,20], on the other hand Pb substitution induced an enhancement of $T_c$. Therefore, the effect of substitution on superconductivity does not correlate to the strength of the scattering by the substituted atoms.

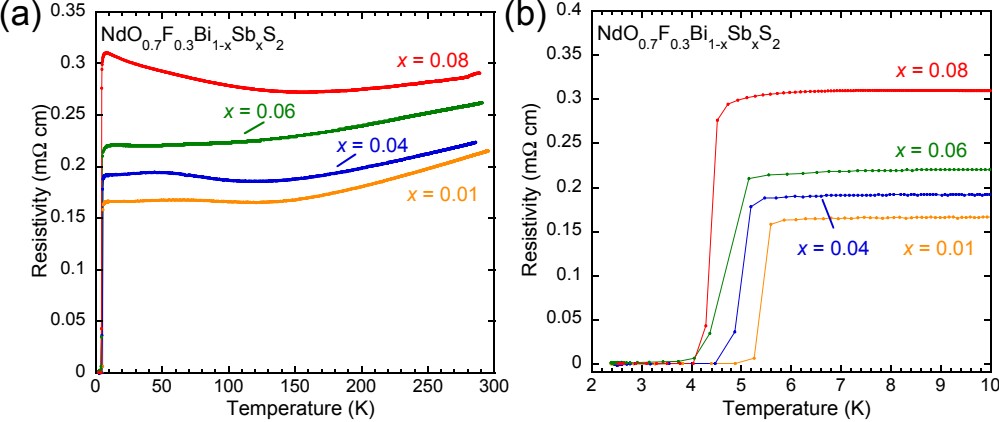

**Figure 4.** Temperature dependence of electrical resistivity for NdO$_{0.7}$F$_{0.3}$Bi$_{1-x}$Sb$_x$S$_2$. (**a**) Temperature dependence of electrical resistivity for NdO$_{0.7}$F$_{0.3}$Bi$_{1-x}$Sb$_x$S$_2$ ($x$ = 0.01–0.08) from 2 to 300 K. (**b**) Temperature dependence of electrical resistivity for NdO$_{0.7}$F$_{0.3}$Bi$_{1-x}$Sb$_x$S$_2$ ($x$ = 0.01–0.08) from 2 to 10 K.

Figure 5a shows a summary of the observed $T_c$. Here, $T_c^{mag}$ is defined as the temperature at which magnetic susceptibility begins to decrease. $T_c^{onset}$ and $T_c^{zero}$ are defined as the temperature at which resistivity begins to decrease and the temperature where resistivity becomes zero, respectively. All $T_c$ values monotonically decrease with an increase in Sb concentration.

Mizuguchi et al. reported that the $T_c$ of Bi$Ch_2$ superconductors is correlated with in-plane chemical pressure [17]. To compare the result in Sb-substituted samples with the previous report, we estimated the in-plane chemical pressure for NdO$_{0.7}$F$_{0.3}$Bi$_{1-x}$Sb$_x$S$_2$. In-plane chemical pressure is defined by the ionic radius of the Bi and chalcogenide ions, as well as the distance between the Bi and chalcogenide ions located in the Bi$Ch_2$ plane. In the case of Sb substitution, this in-plane chemical pressure is defined as follows:

$$\text{in-plane chemical pressure} = ((1 - x)R_{Bi} + xR_{Sb} + R_{Ch})/\text{Bi-}Ch_1 \text{ (in-plane) distance}$$

where $R_{Bi}$ is the ionic radius of the Bi ion. In this study, the ionic radius of the Bi ion was obtained from the structural data of the LaO$_{0.77}$F$_{0.23}$BiS$_2$ single crystal [25]. Thus, the value of $R_{Bi}$ was 104.75 pm, which is almost the same value as that in the previous report [17]. $R_{Sb}$ and $R_{Ch}$ are the ionic radii of Sb$^{3+}$ and $Ch^{2-}$ ions, respectively. These values were determined as 76 and 184 pm, based on [26]. The composition of the Sb ion was used as the nominal value because this value in [17] was also calculated by using the nominal composition of the substituted ions. The Bi–$Ch_1$ distance in this study was estimated by the lattice constant along the $a$-axis. Since the lattice constant along the $a$-axis corresponds to the length of a side of the Bi square lattice, we used $1/\sqrt{2}$ times the lattice constant along the $a$-axis as the Bi–$Ch_1$ distance. Figure 5b shows the estimated in-plane chemical pressure dependence of $T_c^{onset}$, $T_c^{mag}$, and $T_c^{zero}$, shown in open circles, filled circles, and squares, respectively. The $T_c$ of all samples decreases with the decrease in in-plane chemical pressure. Based on this result,

$T_c$ in the Sb-substituted materials is positively correlated to the in-plane chemical pressure. To compare this result with the results with respect to $Ce_yNd_{1-y}O_{0.5}F_{0.5}BiS_2$ and $Nd_ySm_{1-y}O_{0.5}F_{0.5}BiS_2$ in [17], the $T_c$ of $Ce_yNd_{1-y}O_{0.5}F_{0.5}BiS_2$ and $Nd_ySm_{1-y}O_{0.5}F_{0.5}BiS_2$ are superimposed in Figure 5b, which are shown in filled triangles. The $T_c$ in these materials shows a same change as our results against the in-plane chemical pressure. Thus, our result is qualitatively consistent with the previous results of [17]. In addition, these results suggest that the concept of in-plane chemical pressure is valid in the case of substitution not only at the *Ln* site in the blocking layer but also at the Bi site in the conduction layer.

In this study, the in-plane chemical pressure was estimated by the lattice constant along the *a*-axis. On the other hand, the in-plane chemical pressure in [17] was estimated by the Bi–$Ch_1$ distance, which was obtained from single crystal analysis. To compare the difference between these estimated in-plane chemical pressure values, both values were estimated using data from the single crystal analysis in [23]. In Figure 5b, the in-plane chemical pressure estimated by the Bi–$Ch_1$ distance is plotted as a triangle outline, while that estimated by the *a*-axis is plotted as an upside-down triangle. The latter, compared with the former, has a slightly higher value. However, since this value is located at the extrapolated line of the $T_c$ of the Sb-substituted samples, this estimation can be used to estimate the in-plane chemical pressure qualitatively.

Finally, we compared this effect of Sb substitution with the effect of Pb substitution in $NdO_{0.7}F_{0.3}BiS_2$ because this Pb ion is expected to be substituted into the Bi site. Sb substitution suppresses superconductivity. This suppression is explained by the decrease in in-plane chemical pressure described in this paper. On the other hand, Pb substitution increased $T_c$ to up to 6 K. The in-plane chemical pressure of Pb-substituted samples is almost constant because an ionic radius of the $Pb^{2+}$ ion is almost the same as that of the $Bi^{2.7+}$ ion calculated by the single crystal analysis. The ionic radius of the $Pb^{2+}$ ion is 100 pm [26]. If the Pb ion is substituted until $x = 0.10$, $R_{Bi} + R_{Pb}$ is around 103.8 pm, which is almost the same value of the ionic radius of the $Bi^{2.7+}$ ion of 104.75 pm. This indicates that the $T_c$ of Pb-substituted samples is independent of the value of the in-plane chemical pressure.

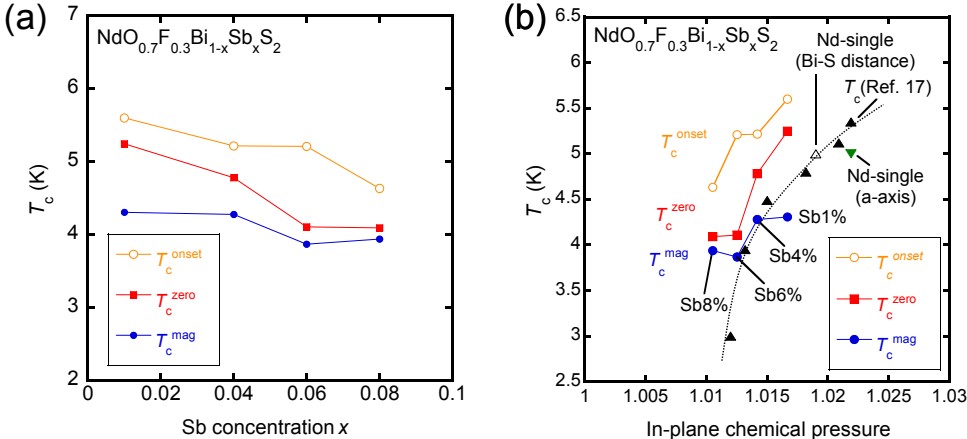

**Figure 5.** (**a**) Sb concentration $x$ dependence of $T_c$ for $NdO_{0.7}F_{0.3}Bi_{1-x}Sb_xS_2$. (**b**) Estimated in-plane chemical pressure dependence of $T_c$ for $NdO_{0.7}F_{0.3}Bi_{1-x}Sb_xS_2$. The $T_c$ values of $(Ce,Nd)O_{0.5}F_{0.5}BiS_2$ and $(Nd,Sm)O_{0.5}F_{0.5}BiS_2$ are plotted as triangles, which were fitted by hand on the dotted line [18].

## 4. Conclusions

An examination of Sb substitution of less than 8% was performed with $NdO_{0.7}F_{0.3}BiS_2$. The Sb substitution caused a decrease in lattice constants along the *c*-axis, as well as a decrease in $T_c$ and the superconducting volume fraction. The deterioration by these superconducting properties by the partial Sb substitution into the Bi site is explained in terms of the decrease in in-plane chemical pressure, indicating the utility of in-plane chemical pressure in Bi$Ch_2$ superconductors.

**Acknowledgments:** This work was partly supported by a Grant-in-Aid for Young Scientists (B) (No. 15K17710) and the Nanotech Career-up Alliance (Nanotech CUPAL).

**Author Contributions:** Satoshi Otsuki, Satoshi Demura and Hideaki Sakata conceived and designed the experiments; Satoshi Otsuki mainly performed the experiments; Satoshi Otsuki, Satoshi Demura and Yuita Fujisawa contributed the data analysis; Yoshihiko Takano contributed analysis tools; Satoshi Demura wrote the paper. All authors have read and approved the final manuscript.

**Conflicts of Interest:** The authors declare no conflict of interest.

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
