# Peer review of "Single Crystal Growth and Superconducting Properties of Antimony-Substituted NdO0.7F0.3BiS2"

_condensedmatter, doi:10.3390/condmat3010001_

Reviewer 1 Report

I found the MS "Single crystal growth and superconducting properties of Antimony Substituted NdO0.7F0.3BiS2 by Satoshi Demura et al interesting. 

The single crystals are always welcome of any new material in comparison to bulk. The superconductivity of bulk BiS2 systems was discovered recently by Mizuguchi group. I am surprised to note, another very quick contributor to this new class of superconductors is completely missing from references list... following articles need to be included in list of refs before publication... there are other mistakes as well.

1. ref 1. Phys. Rev. B 86 (2012) 220510(R). is for Bi4O4S3 and NOT  for Ln(O,F)BiS2 (Ln=La, Ce, Pr, Nd, Sm, Yb and Bi)..... The first sentence of the introduction must be cghanged.... the BiS2 based sup. was discovered in Bi4O4S3 [1,2].. 1. PRB 86(2012)220510(R), 2. JACS 134 (2012) 16504. 

2. For Ln(O,F)BiS2 (Ln=La, Ce, Pr, Nd, Sm, Yb and Bi)... the J. Sup. Novel. mag. 26 (2013) 499, Solid State Commun. 157 (2013) 31-33 and J. Appl. Phys. 113, 056102 (2013) must be cited. 

3. For pressure effect on Tc of BiS2 based ones... the J. Phys. Soc. Jap. Vol. 83, 063707 (2014) must be included. 

So, please arrange the references accordingly... also include an important article appeared today on arxiv.. i.e.  arXiv:1710.10841

After all above corrections the MS can be published. 

Author Response

I attached the reply below. Please see the file.

Reviewer 2 Report

The manuscript "Single crystal growth and superconducting properties of Antimony Substituted NdO0.7F0.3BiS2" by S. Demura and co-authors shows an interesting approach for controlling superconducting response by modification of  in plane chemical pressure in NdO0.7F0.3BiS2 by partial substitution of Bi with  Sb.  The present form  of the paper needs to be improved  before publication. The suggested changes are listed below:

1.       In the manuscript there is no information about a method used for structural data refinement, what were the goodness of fit parameters for  the substituted crystals? The information in the figure caption of Fig.1  should contain information about the form of measured material.

2.       How the authors judge the crystal quality? The width of reflections given in the Fig. 1 is quite large, which raises questions about their homogeneity:  XRF or EDX mapping would surely give more information about the exact nature of these crystals.. The addition of graph with  rocking curves showing crystal quality would be very helpful here.  

3.       There is no information about the exact  samples compositions determined i.e by  any independent method (i.e. SEM-EDX ). The nominal compositions, that are shown in the paper do not give a full picture here because the real composition of the crystal may differ significantly from the nominal one.  ,

4.       The SEM or optical images of the obtained crystals will be also helpful to show their dimensions and morphology.

5.       Another evidence, which suggests the inhomogeneity of the obtained crystals is the large width of superconducting transitions in ZFC mode shown in Figure 2. Please comment on it.

Author Response

(The authors gave the same response as above.)

Reviewer 3 Report

attached below

Author Response

I attached the reply below. Please see the file.

Round  2

Reviewer 1 Report

The MS can be published now.

Author Response

Dear Reviewer,

Thank you for your precious time. I do appreciate for your kind suggestion.

Reviewer 2 Report

condensedmatter-240855v2

The manuscript "Single crystal growth and superconducting properties of Antimony Substituted NdO0.7F0.3BiS2" by S. Demura and co-authors has been considerably improved and the Authors answered to most of the raised questions. However the manuscript still needs some minor corrections, that are listed below:

1) The elemental distribution maps for the grown crystals, which will show their homogeneity are still missing.  The authors collected electron microscopy images therefore,  it would be possible to add these maps to the current paper as well.

2)  Please check spelling in the whole manuscript as there are still some misprints, i.e.: at p. line 120 in the sentence: " Therefore, the miner region..."  it should be: "Therefore, the minor region..." 

3) The answers given for points 3&4, related to the crystals homogeneity  and their real are not sufficient.

"Answer to comment 3 and 4:

I performed the composition analysis by EDX for single crystals shown in the Fig. A2. Although the Sb ion is detected by the EDX measurement, the concentration of Sb ion only changed between 1-3 %. The Sb concentration in the sample is smaller than that nominal value. However, the Sb concentration is not determined quantitatively because the resolution of EDX is low to measure the difference of the Sb concentration of 1 %."

With EDX technique it s possible to detect these differences as typical detection limits are about 1000 ppm (0.1% by weight), even smaller contents can detected by using longer counting times.

"Therefore, I took the backscattered electron images to evaluate the homogeneity of the Sb concentration as show in Fig. A3. From these images, the homogeneous surface is seen in these single crystals. The surface of x=0.08 is slight dirty because some dust is added on the surface. On the other hand, the clean surface, which is the NdO0.7F0.3Bi1-xSbxS2, is quite homogeneous. This indicates the spatial distribution of Sb ion is quite small within the resolution of EDX measurement"

According to the first part of the answer the detected amounts of Sb are far from the nominal composition , therefore it is crucial to see the distribution of this element in the material. Backscattered electron images are not sufficient here. First of all they were done with too small magnification and concluding on the crystal homogeneity and uniform distribution of all elements from them is rather controversial.

4) The SEM images shown in the Responses should be also included to the amended version of manuscript. The same applies to EDX maps showing  elemental distributions for: Nd, O, F,  Bi, Sb and S.

Author Response

Dear Reviewer,

I do appreciate for the kind suggestion. Since I agree with your suggestions, the manuscript was modified.Please check the attach PDF file.

Reviewer 3 Report

I am happy to see that all of referee comments are addressed in the revised manuscript.

I believe that the manuscript deserves to be published in Condensed Matter in the present form.

Author Response

(The authors gave the same response as above.)
